# Validation of a Video-Based Performance Analysis System (Mediacoach^®^) to Analyze the Physical Demands during Matches in LaLiga

**DOI:** 10.3390/s19194113

**Published:** 2019-09-23

**Authors:** Jose Luis Felipe, Jorge Garcia-Unanue, David Viejo-Romero, Archit Navandar, Javier Sánchez-Sánchez

**Affiliations:** 1Universidad Europea de Madrid, Faculty of Sport Sciences, Villaviciosa de Odón, 28670 Madrid, Spain; joseluis.felipe@universidadeuropea.es (J.L.F.); david.viejo@universidadeuropea.es (D.V.-R.); archit.navandar@universidadeuropea.es (A.N.); 2Universidad de Castilla-La Mancha, IGOID Research Group, 45071 Toledo, Spain; jorge.garciaunanue@uclm.es

**Keywords:** time-motion analysis, GPS, match-tracking, football, physical demands

## Abstract

The aim of the present study was to assess the accuracy of a multi-camera tracking system (Mediacoach^®^) to track elite football players’ movements in real time. A total of 207 observations of 38 official matches from Liga 1, 2, 3™ (2nd Spanish Division, season 2017/18) were included in the study (88 defenders, 84 midfielders, and 35 attackers of the same team). Total distance (TD, m) distance in zone 4 (DZ4) at a speed of 14–21 km/h, distance in zone 5 (DZ5) at a speed of 21–24 km/h (DZ5), distance in zone 6 (DZ6) at a speed of ≥24 km/h, maximum speed (km/h), and number of sprints (actions above 24 km/h) were registered with the Apex^®^ GPS system (STATSports™, Newry, N. Ireland) and Mediacoach^®^ semi-automatic tracking system (LaLiga™, Madrid, Spain). The level of agreement between variables estimated by the two systems was analyzed. Bias was also calculated by deducting the GPS estimated value from the video estimated value, and then dividing the difference score by the GPS estimated value. All variables showed high ICC values (>0.75) and very large correlations (r > 0.70). However the video-based performance analysis system overestimated the results obtained in the different speed zones (DZ5: +16.59 ± 62.29 m; LOA95%: −105.49 to 138.68; DZ6: +93.26 ± 67.76 m; LOA95%: −39.55 to 226.07), the number of sprints (+2.27 ± 2.94; LOA95%: −3.49 to 8.02), and the maximum speed (+0.32 ± 1.25 km/h; LOA95%: −2.13 to 2.77). The maximum bias was found in DZ6 (47%). This demonstrates that Mediacoach^®^ is as accurate as a GPS system to obtain objective data in real time, adapted to physical and movement demands of elite football, especially for total distance and distances traveled at medium speeds.

## 1. Introduction

Match analyses in football give more knowledge about the physiological and technical demands of the game [1]. These studies provide very useful information about players’ performances during the entire match [2] and during different periods of the game [3]. On average, a player covers 9–11 km per game [4], but this varies based on specific positions [5]. 

However, the actions of the players on the football pitch are intermittent in nature [6], consisting of irregular and complex patterns that involve accelerations and decelerations in different directions and over short periods of time [7]. Such intermittent actions can be split into different intervals, using velocity thresholds: high-intensity runs (14–21 km/h), very high-intensity runs (21.1–24 km/h), and sprints (over 24 km/h). The distances run at higher velocities are progressively lower as the velocity ranges increase [8]: 1614 ± 320 m for high-intensity runs [9], 847 ± 349 m for very high-intensity runs [10], and 184 ± 87 m for sprints [11]. Research in performance analysis of football suggests that the reaching of peak velocities and the distance covered at very high intensities are crucial [12]; hence, variables, such as the number of sprints in a game and the peak velocity reached by an outfield player are crucial in analyzing match performance data. The number of sprints varied between 7 to 61 sprints per game [4], and this varied based on player position and playing half [5], with players attaining maximum velocities of 31.9 ± 2.0 km·h^−1^ [13]. Precise knowledge about match performance data is extremely important for members of the technical staff of different teams, as they use this information to design training sessions accordingly. 

Contemporary tracking technologies help monitor physiological and physical loads during matches, and this in turn helps adapt training sessions according to the match demands [9]. Among these systems, the Global Positioning System (GPS) is considered a valid instrument to collect physiological parameters in elite football [12] during training and matches, with computerized semi-automatic tracking systems also being used to collect physical performance variables [1]. However, different professional leagues use different tracking systems, and not all tracking systems were validated against a gold-standard instrument [1]. Since 2011, the Spanish Professional Football League (Liga de Futbol Profesional, LaLiga™) has used the Mediacoach^®^ system (LaLiga™, Madrid, Spain) to provide semi-automatic tracking of players for all matches in the first and second divisions. There are limited match-analysis studies performed with this system [5], and this system is not yet validated against a gold-standard instrument. 

This validation is very important, considering that Spanish clubs have dominated continental and European football over the last decade [14]; and playing tactics vary based on leagues [3] and, as a consequence, so do the match performance data [15]. Hence, the aim of the present study was to assess the accuracy of the multi-camera tracking system (Mediacoach^®^) to track elite football players’ movement in real time. 

## 2. Materials and Methods

### 2.1. Participants

A total of 24 professional football players from the same team took part in the study (26 ± 6.5 years of age). Players were classified as defenders, midfielders, and attackers. All the players were notified of the research design and its requirements, as well as the potential benefits and risks, and this research was approved by the Health Sciences Research Committee of the European University of Madrid (CIPI045/19), based on the latest version of the Declaration of Helsinki.

### 2.2. Experimental Design

The criterion validity of the Mediacoach^®^ multi-camera tracking system was evaluated by the concurrent validity of different measures of distance, sprints, and velocity between the video system and GPS system in a professional Spanish Football League (LaLiga™). A total of 38 official matches were registered from Liga 1, 2, 3™ (2nd Spanish Division) that were played out throughout the 2017–2018 season by a professional team. Seven field players of the starting team were randomly selected at the beginning of each match and equipped with a GPS (the team had only that GPS number, which was used only in official matches). In total, 266 observations were collected. To allow data comparison between both systems, only observations that recorded all minutes played from both systems were used. Therefore, 59 records were eliminated due to disconnection or signal-lost problems in the GPS device during the game. Finally, a total of 207 observations were included in the study (88 defenders, 84 midfielders, and 35 attackers). Each player in each match was identified, and the full data set from both systems was obtained, so the final concurrent data was synchronized (Figure 1). Of the 207 records, in 128 of them, the player completed the entire match. In the rest (79), the player was replaced during the match, although both systems collected information on all minutes played. The average time recorded was 77 ± 27 min.

The GPS system used to contrast the measurements obtained by the video system in the same matches was Apex^®^ GPS system (STATSports™, Newry, N. Ireland), which was previously validated [16]. Apex^®^ GPS 10 Hz presents the following characteristics: dimensions of 30 mm (wide) × 80 mm (high), a weight of 48 g, a 100 Hz gyroscope, a 100 Hz tri-axial accelerometer, and a 10 Hz magnetometer. Apex^®^ GPS 10 Hz showed distance bias of 1.05 ± 0.87%, 2.3 ± 1.1%, and 1.11 ± 0.99% in the 400 m trial, 128.5 m circuit, and 20 m trial, respectively, and a Vpeak bias of 26.5 ± 2.3 km h^−1^ [16]. Apex^®^ 10 Hz is a multi-GNSS (Global Navigation Satellite Systems) augmented unit, which is capable of acquiring and tracking multiple satellite systems (e.g., GPS, GLONASS, Galileo, BeiDou). Global Navigation Satellite Systems data (speed and distance) recorded by the units were downloaded and further analyzed by the STATSports™ Apex^®^ Software (Apex^®^ 10 Hz version 2.0.2.4) [16].

Data obtained were compared with those offered by the Mediacoach^®^ multi-camera tracking system (LaLiga™, Madrid, Spain). Mediacoach^®^ is a multi-camera tracking system that has a frequency of 25 Hz. To collect data, 8 different cameras were situated strategically in order to follow and track the 22 players on the field throughout the match. Previously, other similar systems were also validated to determine the physical activity of the players in competition, such as Prozone^®^ [17], Amisco Pro^®^ [8], or Venatrack^®^ [18]. Mediacoach^®^ records from several angles and analyzes X and Y positions for each player, resulting in three-dimensional tracking in real time (tracking data were recorded at 25 Hz per second). Mediacoach^®^ is also based on data correction of the semi-automatic video technology (the manual part of the process). This correction is made by an overlay of the X coordinate, provided automatically by the system for each player on the real video image of the match. This detects and visually corrects the situations in which the positioning coordinates are erroneous because they move away from the position of the player to whom the data belong [19].

The same variables were obtained from both systems: total distance (TD, distance covered by the player throughout the match independent from the speed at which players were moving); distance in zone 4 (DZ4, distance covered by the player at a speed greater that 14 km/h and lower than 21 km/h), distance in zone 5 (DZ5, distance covered by the player at a speed greater that 21 km/h and lower than 24 km/h), distance in zone 6 (DZ6, distance covered by the player at a speed equal to or greater than 24 km/h), maximum speed (Vmax, peak height velocity achieved by the player recorded in km/h), and number of sprints (NSprint, number of actions above 24 km/h). 

### 2.3. Statistical Analysis

Mediacoach^®^ data are provided directly by the provider (LaLiga™), with reports per game for all clubs. GPS data are downloaded in a dock station and filtered with GPSport^®^ software. The level of agreement between the variables estimated by the video system and by the GPS devices was analyzed. The following tests were performed: mean error, mean absolute error (MAE), root mean square error (RMSE), product moment correlation (Pearson’s r), intraclass correlation coefficient (ICC) of the total agreement, and standard error of the estimate in standardized terms (SEE). Bias was also calculated by deducting the GPS estimated value from the video estimated value, and then dividing the difference score by the GPS estimated value. Finally, Bland–Altman plots with mean error and 95% limits of agreement were included. Product moment correlations results were evaluated as trivial (<0.10), small (0.10–0.30), moderate (0.30–0.50), large (0.50–0.70), very large (0.70–0.90), and nearly perfect (0.90–1.00) [20]. ICC results were evaluated as very low (<0.20), low (0.20–0.50), moderate (0.50–0.75), high (0.75–0.90), very high (0.90–0.99), and extremely high (>0.99) scores [21]. SEE was interpreted as trivial (<0.10), small (0.10–0.30), moderate (0.30–0.60), large (0.60–1.00), very large (1.00–2.00), and extremely large (>2.00), using Hopkins’s reliability spreadsheet [22]. All calculations were carried out in SPSS v20.0 (IBM Corp. Released 2011. IBM SPSS Statistics for Windows, Version 20.0. Armonk, NY: IBM Corp) and Stata v14.0 (StataCorp. 2015. Stata Statistical Software: Release 14. College Station, TX: StataCorp LP.).

## 3. Results

Total distance shows the highest similarities between Mediacoach^®^ and GPS (Table 1), with nearly perfect correlations (r = 0.99) and very high ICC (0.99). The remaining variables showed high ICC values (>0.75) and very large correlations (r > 0.70). According to SEE, the results show a small (TD (0.14) and DZ4 (0.28)) or moderate (DZ5 (0.53), DZ6 (0.40) and NSprint (0.39)) errors between systems, except for Vmax with a large SEE (0.70).

Plots obtained through the Bland–Altman analysis are presented in Figure 2. Results show deviations between the video-based performance analysis system and GPS devices in TD (−2.17 ± 429.03 m; LOA95%: −843.07 to 838.72). According to the distances covered in different speed zones, the video-based performance analysis system overestimated the results obtained, and the overestimation was lower in high-intensity speed zones (D_Z5_: +16.59 ± 62.29 m; LOA95%: −105.49 to 138.68; D_Z6_: +93.26 ± 67.76 m; LOA95%: −39.55 to 226.07). Finally, the Bland–Altman test revealed a slight overestimation in the number of sprints (+2.27 ± 2.94; LOA95%: −3.49 to 8.02) and maximum speed (+0.32 ± 1.25 km/h; LOA95%: −2.13 to 2.77) when obtained with Mediacoach^®^ system.

## 4. Discussion

This paper aimed to validate the Mediacoach^®^ multi-camera tracking system against an Apex^®^ GPS system at 10 Hz. This is pioneering research, showing the accuracy of a multi-camera system using match performance data in elite football. The study found a good concordance between both measurement systems, maintaining a very constant and homogeneous variation in the measurements.

The correlations between distances and participants recorded via Mediacoach^®^ multi-camera tracking system and the GPS were all strong (r > 0.80), including very strong correlations (r > 0.95) in the TD and DZ4. Similar correlations were detected in the validation of other multi-camera tracking systems, such as Prozone^®^ [17] and Venatrack^®^ [18]. In addition, high ICC values were found for all variables (ICC ≥ 0.75), including a very high value of TD (ICC = 0.99). These results are in line with those detected in previous studies [8,20]. Small (<0.30) and moderate SEE (<0.60) were found for all cases except Vmax (SEE = 0.70). The main finding of this study is that the video system tends to record higher values in all physical variables analyzed, although it is very consistent in all of its measurements, so it is very easy to predict the measurement error. This aspect was detected by Buchheit et al. [20] in Prozone^®^ with U14–U17 players, proposing a constant correction factor to solve the constant error. In addition, the differences found between both systems are in line with those found between different GPS systems and dual-beam timing gates (the gold standard) [19]. From these publications, we can deduce that, in general, GPS systems underestimate the distance traveled.

Previous investigations [23] found accurate accuracy values of GPS versus dual-beam timing gates (1–1.13% error) in very short distances (10–20 m sprint). However, as the distance increases (TD) and the speeds are lowered (DZ4), the error rate decreases further, making the GPS systems much more accurate. Waldron, Worsfold, Twist, and Lamb [24] explained that the accuracy of GPS measurement systems decreases as distance and speed increase, since players in these types of situations deviate from the linearity of sprint, where the systems of time gate methods were declared more efficient. Thereby, we can deduce that this multi-camera tracking system has a high level of accuracy in real situations of elite football, taking into account the nonlinear displacements of the players.

In the same way, data found in this study show lower measurements recorded by GPS when the speed increases (DZ5; DZ6) and less NSprints. These same results were presented in previous studies for different team sports [25,26], for which GPS devices have an acceptable accuracy and validity to evaluate the total distance during exercises and longer-duration training games. However, they cannot be used to measure brief high-intensity sprints or short and rapid accelerations at distances of less than 20 m, irrespective of the sampling frequency, since movement changes limit the opportunity to record positional measurements [24]. Therefore, the greatest discrepancies found in higher-distance speeds should not be attributed solely to the video system, since the literature indicates a lower validity of the GPS system in this zone.

The differences found in the measurements of the number of sprints between both systems can be explained by a variety of factors, all of which are related to the accelerometer incorporated in the GPS systems. Firstly, the measurements of these accelerometers can be influenced by their placement [27] and body-orientation changes [28]. As velocities increase from high intensities to sprint velocities, there is an increased trunk lean [29], which can generate an over-quantification in the number of sprints [24]. Secondly, other factors, such as the GPS vest supplied by the manufacturer or the external noise in the stadiums [30], have a decisive influence on the accuracy of the quantification of sprints.

Regarding the maximum speeds, differences were found with moderate SEE levels (SEE = 0.70, r = 0.81), with this being the variable with the least precision between both systems. The differences may be due to the low precision of GPS sensors in quantifying high-speed movements and accelerations over short distances [31]. This error can invalidate the use of this type of system to quantify the maximum speed in team sports, since it is considered as a key performance indicator for sports such as football, rugby, or hockey [32,33,34].

Regarding the data divided by positions, the Mediacoach^®^ system measurements for forwards and midfielders were more accurate, with SEE for all the variables being moderate (SEE < 0.50) and a high correlation with GPS measurements (r > 0.86). Further distortions for the group of defenders were found. This can be explained by the greater TD covered by them (10307.33 ± 1206.33) as compared to forwards (7240.61 ± 3411.31), midfielders (7705.06 ± 3201.10), or the average of all players irrespective of position (8622.87 ± 3021.42). When greater distances are found, the probability of error is increased. However, the adjustment that is made for each of the variables analyzed by each system predicts in a very high way the results that can be obtained with the measurements by the other system of this research.

One of the limitations of this study was that the players were grouped together in three broad categories: defenders, midfielders, and forwards, given the small sample size. For example, in the case of defenders, the central and lateral defenders have different technical and physiological characteristics, which could affect the results. Another limitation was that not all players who participated in the match wore the vest with the GPS sensors, as a personal decision, hence the number of cases that were compared was also restricted. Collecting data over a longer period, with more players, could help overcome these limitations. Finally, it must be taken into account that GPS systems are not considered gold standard, so the error associated with this device is able to affect the results. Previous studies showed some errors around 1–2% in GPS; therefore, this metric variation among players and training sections should be analyzed with these errors in mind [35].

## 5. Conclusions

The large correlation and regression coefficients found, together with the small SEE values for all the variables in this study, and for each of the analyzed positions, not only show that Mediacoach^®^ is as reliable and valid as a GPS system, but also demonstrates a high accuracy in tracking the specific movements of elite football. Finally, it is clear that this system could be used by coaches, strength and conditioning coaches, sports scientists, and even by the players to obtain valid objective data in real time, without the need for players to incorporate a device on their backs, with the inconvenience that this may cause.

## Figures and Tables

**Figure 1 sensors-19-04113-f001:**
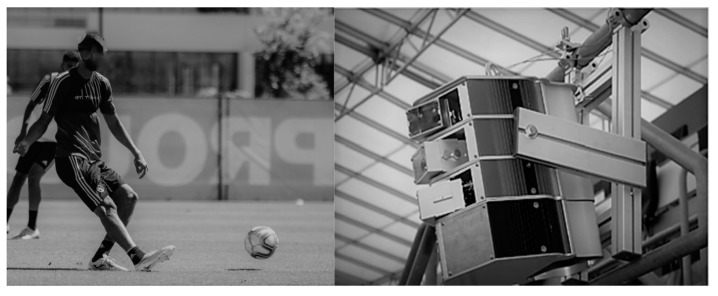
GPS system placed on a player (**left**) and Mediacoach^®^ cameras in a stadium (**right**).

**Figure 2 sensors-19-04113-f002:**
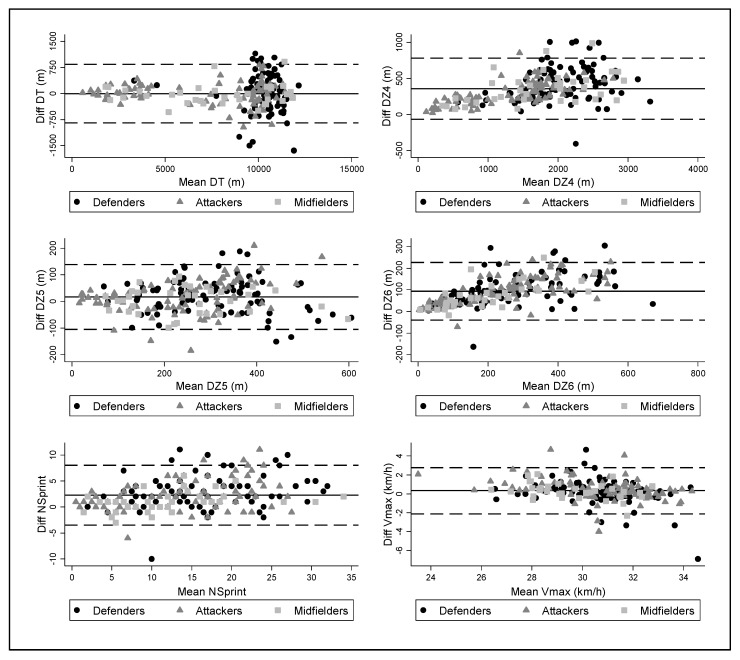
Bland–Altman plots identifying differences when comparing measurements with the Global Positioning System (GPS) devices and the video-based performance analysis system (Mediacoach^®^). Central line represents the inter-methods difference. Upper and lower lines represent the 95% limits of agreement (mean differences ± 1.96 SD of the differences).

**Table 1 sensors-19-04113-t001:** Validity for the distances, velocity, and number of sprint measurements recorded by both the Global Positioning System (GPS) devices and Mediacoach^®^ during a match, according to the playing position (n = 207).

		Mediacoach^®^	GPS	Diff	MAE	RMSE	r	ICC	SEE	Bias
Total (n = 207)	TD	8622.87	8625.04	−2.17	313.41	427.99	0.99	0.99	0.14	0.00
		(3021.42)	(3032.47)							
	D_Z4_	1871.99	1512.08	359.91	363.85	419.80	0.96	0.84	0.28	0.24
		(747.11)	(643.38)							
	D_Z5_	276.36	259.76	16.59	50.71	64.31	0.88	0.88	0.53	0.06
		(131.74)	(126.34)							
	D_Z6_	291.96	198.70	93.26	95.87	115.18	0.93	0.75	0.40	0.47
		(165.84)	(126.22)							
	N_Sprint_	15.48	13.22	2.27	2.75	3.70	0.93	0.89	0.39	0.17
		(8.09)	(7.13)							
	V_max_	30.69	30.37	0.32	0.86	1.29	0.81	0.78	0.70	0.01
		(1.73)	(2.12)							
Defenders (n = 88)	TD	10,307.33	10,301.94	5.39	419.84	545.45	0.90	0.90	0.48	0.00
		(1206.45)	(1251.03)							
	D_Z4_	2242.22	1823.68	418.55	427.81	485.20	0.89	0.67	0.50	0.23
		(550.70)	(498.31)							
	D_Z5_	313.61	291.75	21.86	54.65	68.60	0.85	0.84	0.61	0.07
		(116.02)	(124.78)							
	D_Z6_	313.44	214.89	98.55	102.26	121.97	0.90	0.71	0.49	0.46
		(160.37)	(127.80)							
	N_Sprint_	16.81	14.25	2.56	3.01	4.04	0.92	0.86	0.44	0.18
		(7.80)	(7.01)							
	V_max_	30.69	30.49	0.20	0.90	1.42	0.68	0.65	1.09	0.01
		(1.45)	(1.90)							
Attackers (n = 84)	TD	7240.61	7240.75	−0.14	243.45	323.18	1.00	1.00	0.10	0.00
		(3411.31)	(3431.80)							
	D_Z4_	1481.91	1183.25	298.66	298.66	338.45	0.98	0.88	0.19	0.25
		(697.72)	(588.88)							
	D_Z5_	246.76	230.31	16.46	50.08	65.20	0.89	0.88	0.50	0.07
		(141.40)	(120.50)							
	D_Z6_	293.58	197.77	95.81	97.93	115.73	0.95	0.76	0.34	0.48
		(172.78)	(127.89)							
	N_Sprint_	15.19	12.73	2.46	2.92	3.83	0.94	0.88	0.39	0.19
		(8.22)	(7.14)							
	V_max_	31.05	30.60	0.45	0.89	1.28	0.86	0.83	0.59	0.01
		(1.94)	(2.35)							
Midfielders (n = 35)	TD	7705.06	7731.13	−26.08	213.72	290.99	1.00	1.00	0.09	0.00
		(3201.10)	(3182.13)							
	D_Z4_	1877.31	1517.84	359.48	359.48	418.89	0.97	0.87	0.26	0.24
		(828.71)	(728.22)							
	D_Z5_	253.72	250.05	3.67	42.29	49.30	0.92	0.92	0.42	0.01
		(124.98)	(128.59)							
	D_Z6_	234.09	160.25	73.85	74.84	94.42	0.94	0.78	0.35	0.46
		(152.65)	(112.25)							
	N_Sprint_	12.86	11.80	1.06	1.69	2.21	0.97	0.96	0.25	0.09
		(7.97)	(7.23)							
	V_max_	29.81	29.52	0.29	0.68	0.90	0.89	0.87	0.51	0.01
		(1.58)	(1.89)							

TD: total distance; DZ4: distance in zone 4 (14–21 km/h); DZ5: distance in zone 5 (21–24 km/h); DZ6: distance in zone 6 (>24 km/h); NSprint: number of sprints, actions above 24 km/h; Vmax: maximum speed in km/h.

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
