# Peer review of "Validation of a Video-Based Performance Analysis System (Mediacoach®) to Analyze the Physical Demands during Matches in LaLiga"

_sensors, 2019, doi:10.3390/s19194113_

Round 1

Reviewer 1 Report

Dear Autors,

I just want to congratulate you on very interesting research.  It is well writter and presented.

Just one thing, I would like to ask you to check, I think you have one ''using'' too much in line 102

Author Response

Authors would like to thank the reviewer for their work and comments of the manuscript. We have removed the word 'using' from it. 

Kind regards! 

Reviewer 2 Report

I congratulate the authors for the study. It is very necessary and it has an high applicability in the football elite setting. Nevertheless I propose to the authors some suggestions:

1) to add more adequate analysis, Bland and Altman plots and statistics.

2) Furthermore, I think it is necessary to remove all reference to the validity concept. Comparing two tracking system does not give us information about validity but agreement between techniques.

3) I believe is better for implement the analysis to select only the players played the complete match, because if not, it is very easy get a very good correlation between systems.  

Thanks for your effort

Author Response

Authors greatly appreciate the comments and review made.
We have implemented all the changes (including the Bland Altman graphics).
However, we do not eliminate players without a complete match, since we are looking for a comparison between both systems in real game situations and not in the laboratory. In this way, we appreciate this comment, it is very constructive, but we maintain our results

Reviewer 3 Report

Dear authors,

thank you very much for possibility to review and help you to publish your research.  It is relative good and clear described study. The research focused on system verification is very important for future research and practice. Thank you for it.

Author Response

Authors thank this comments. These are very motivating and of great interest for our daily work.